# CRISPR-Cas9 knockdown of ESR1 in preoptic GABA-kisspeptin neurons suppresses the preovulatory surge and estrous cycles in female mice

Jenny Clarkson[1,2], Siew Hoong Yip[1,2], Robert Porteous[1,2], Alexia Kauff[2], Alison K Heather[2], Allan E Herbison[1,2,3]*

[1]Centre for Neuroendocrinology, Dunedin, New Zealand; [2]Department of Physiology, University of Otago School of Biomedical Sciences, Dunedin, New Zealand; [3]Department of Physiology, Development and Neuroscience, University of Cambridge, Cambridge, United Kingdom

*For correspondence:
aeh36@cam.ac.uk

Competing interest: The authors declare that no competing interests exist.

**Abstract** Evidence suggests that estradiol-sensing preoptic area GABA neurons are involved in the preovulatory surge mechanism necessary for ovulation. In vivo CRISPR-Cas9 editing was used to achieve a 60–70% knockdown in estrogen receptor alpha (ESR1) expression by GABA neurons located within the regions of the rostral periventricular area of the third ventricle (RP3V) and medial preoptic nuclei (MPN) in adult female mice. Mice exhibited variable reproductive phenotypes with the only significant finding being mice with bilateral ESR1 deletion in RP3V GABA neurons having reduced cFos expression in gonadotropin-releasing hormone (GnRH) neurons at the time of the surge. One sub-population of RP3V GABA neurons expresses kisspeptin. Re-grouping ESR1-edited mice on the basis of their RP3V kisspeptin expression revealed a highly consistent phenotype; mice with a near-complete loss of kisspeptin immunoreactivity displayed constant estrus and failed to exhibit surge activation but retained pulsatile luteinizing hormone (LH) secretion. These observations demonstrate that ESR1-expressing GABA-kisspeptin neurons in the RP3V are essential for the murine preovulatory LH surge mechanism.

## eLife assessment

This **important** study provides **convincing** evidence of the criticality of estradiol - estrogen receptor-mediated upregulation of kisspeptin within neurons of the preoptic area to generate an ovulation-inducing luteinizing hormone surge. The use of in vivo CRIPSR-Cas9 is novel in this system and provides a road map for future studies in reproductive neuroendocrinology. This paper will be of interest to reproductive neuroscientists and endocrinologists.

## Introduction

Understanding how circulating estradiol levels feedback on the brain to regulate the pulse and surge profiles of gonadotropin-releasing hormone (GnRH) secretion is of central importance to elucidating the neural control of mammalian fertility. Studies in genetic mouse models have clearly identified that estrogen receptor alpha (ESR1) is the key receptor underlying both estrogen-positive and -negative feedback (*Couse et al., 2003*; *Herbison, 2015*). Despite significant progress (*Herbison, 2015*; *Wang et al., 2019*; *Moenter et al., 2020 Goodman et al., 2022*; *Kauffman, 2022*), defining the precise cellular and molecular mechanisms through which ESR1 ultimately

modulates GnRH secretion remains a significant challenge. This is particularly intriguing given that the final output neuron of the network, the GnRH neuron, expresses ESR2 but not ESR1 (*Herbison and Pape, 2001*).

Recent experiments using live cell imaging and in vivo CRISPR gene editing have demonstrated that estradiol acts directly at ESR1-expressing kisspeptin neurons in the arcuate nucleus to suppress the activity of the GnRH pulse generator and bring about estrogen-negative feedback at the level of the brain (*McQuillan et al., 2022*). This same level of definition has not yet been possible for the positive feedback mechanism where only the general location of estradiol action is known with certainty. Viral retrograde labeling demonstrated that ESR1-expressing afferents to the GnRH neuron cell bodies are concentrated in the anteroventral periventricular (AVPV) and preoptic periventricular (PVpo) nuclei of the hypothalamus (*Wintermantel et al., 2006*); collectively termed the rostral periventricular area of the third ventricle (RP3V) (*Herbison, 2008*). As the selective knockdown of ESR1 in the RP3V results in acyclic mice with an absent luteinizing hormone (LH) surge (*Porteous and Herbison, 2019*), it seems very likely that the RP3V is the area within which estradiol acts to enable the surge mechanism in mice.

The identity of the key ESR1-expressing populations within the RP3V mediating the estrogen-positive feedback mechanism has not been defined conclusively. Most evidence supports a role for the RP3V kisspeptin neurons. These cells express ESR1, are activated at the time of the surge, and have their kisspeptin synthesis strongly upregulated by estradiol (*Smith et al., 2006*; *Adachi et al., 2007*; *Clarkson et al., 2008*). Further, RP3V kisspeptin neurons project directly to GnRH neuron cell bodies where they provide a strong and lasting excitatory influence while the selective activation of RP3V kisspeptin neurons in vivo evokes a surge-like increment in LH secretion (*Piet et al., 2018*). Given these findings, it was surprising to find that selective in vivo CRISPR knockdown of ESR1 in AVPV kisspeptin neurons blunted the amplitude of the LH surge but had no impact on estrous cyclicity (*Wang et al., 2019*). This suggested that kisspeptin neurons may not be the key or sole phenotype in the RP3V responsible for positive feedback in mice.

There is a long-standing evidence implicating GABAergic inputs to the GnRH neuron cell bodies in the rodent-positive feedback mechanism. This ranges from studies showing alterations in $GABA_A$ receptor transmission at GnRH neurons around the time of the daily LH surge (*Christian and Moenter, 2007*) to in vivo studies demonstrating a functionally significant fall in GABA release within the preoptic area just prior to the LH surge (*Herbison and Dyer, 1991*; *Jarry et al., 1992*). More recently, the genetic deletion of ESR1 in all GABA neurons was found to result in failure of the positive feedback mechanism (*Cheong et al., 2015*).

The present study was designed to test directly whether ESR1 GABA neurons located in the medial preoptic area are critical for the positive feedback mechanism. We employed in vivo CRISPR gene editing to knock down ESR1 in the RP3V and adjacent medial preoptic nucleus (MPN) and evaluated estrous cyclicity alongside pulsatile and surge patterns of LH secretion.

## Results
### CRISPR knockdown of ESR1 in preoptic area GABA neurons
#### Design and testing of gRNA

The design and characterization of guide RNAs for *Esr1* (NM_007956) have been reported previously (*McQuillan et al., 2022*). In brief, gRNA directed at both the sense and antisense strands around exon 3 were tested for efficacy using an ESR1-expressing hypothalamic cell line mHypo-CLU189-A genetically modified to stably express Cas9 (CLU189-Cas922C). Transduction with AAV-U6-gRNA-EGFP identified gRNA-1, -2, -3, and -6 to be the most effective in reducing *Esr1* mRNA levels by 20–30% in vitro. The efficacy of gRNA-2, -3, and -6 was assessed in vivo by giving unilateral injections of AAV1-U6-gRNA(2, 3, or 6)-Ef1α-mCherry into the medial preoptic area of *Slc32a1^Cre^,Rosa26^LSL-Cas9-EGFP^* (Vgat-Cas9) mice (N=4 per gRNA). This was found to result in 79 ± 7%, 72 ± 5%, and 78 ± 3% reductions, respectively, in the numbers of ESR1-immunoreactive EGFP-expressing neurons on the injected side of the brain compared with the non-injected side. Our prior studies have shown that this CRISPR-Cas9-AAV platform enables the knockdown of ESR1 in a selective manner in genetically targeted hypothalamic kisspeptin neurons (*McQuillan et al., 2022*).

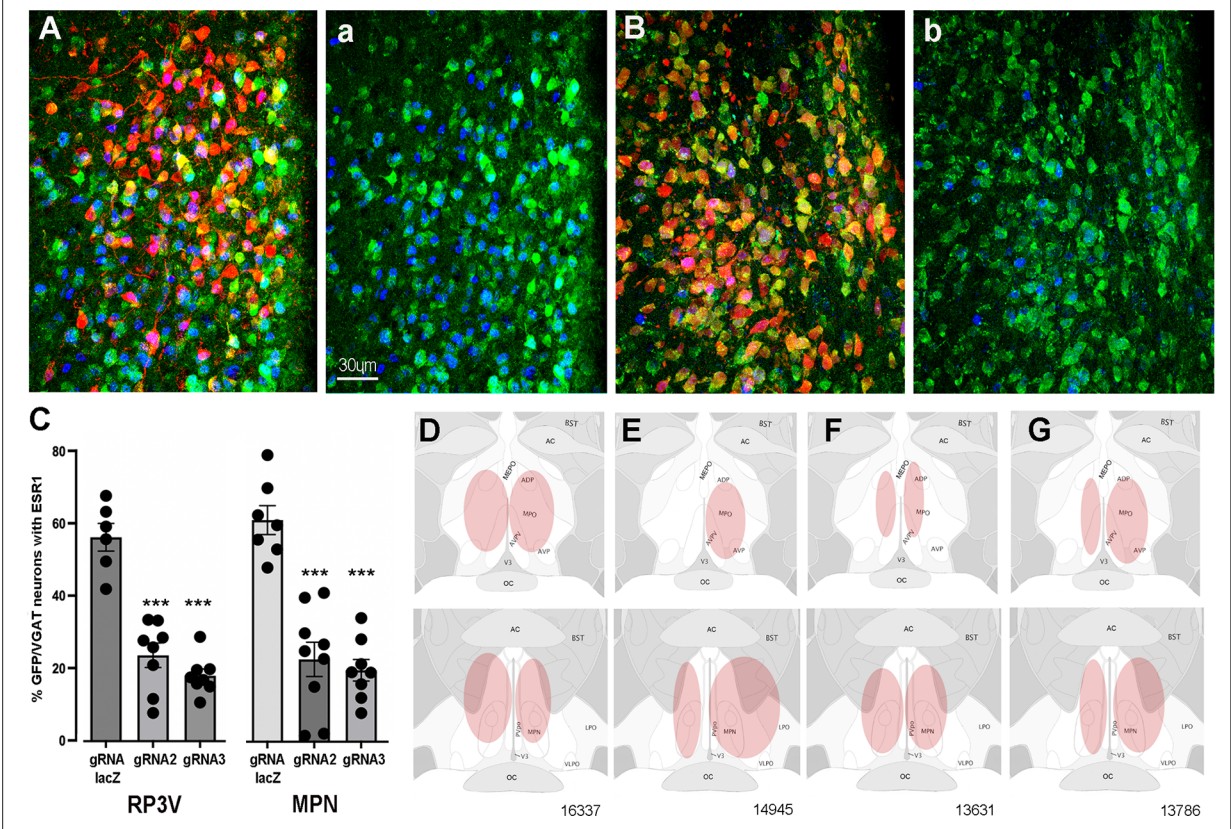

**Figure 1.** CRISPR knockdown of ESR1 in preoptic GABA neurons. (**A, B**) Photomicrographs showing distribution of mCherry (gRNA from AAV) and expression of GFP (Cas9) in VGAT neurons and nuclear-located ESR1 (blue) in the preoptic periventricular (PVpo) of two mice receiving either gRNA-LacZ (**A**) or gRNA-2 (**B**). The mCherry signal is removed in the adjoining plates (a) and (b) so that the VGAT neurons (green) co-expressing ESR1 (blue or teal nuclei) are more easily identified. Scale bar in (a) is the same for all photomicrographs. (**C**) Individual data points (n=6-9) and mean ± SEM percentage of GFP/VGAT neurons expressing ESR1 within injected regions of the RP3V and medial preoptic nuclei (MPN) for the three gRNA groups. ***p<0.001 versus gRNA-LacZ (ANOVA, post hoc Dunnett's tests). Data in *Source data 1*. (**D–G**) Representative schematics of AAV injection sites (pink) in four mice; 16,337 (bilateral gRNA-2), 14,945 (unilateral RP3V, bilateral MPN gRNA-2), 13,631 (bilateral gRNA-3), 13,786 (bilateral gRNA-LacZ).

## ESR1 knockdown in preoptic GABA neurons

In the present study, gRNA-2 and gRNA-3 were used for ESR1 knockdown with their effects in medial preoptic GABA neurons evaluated in Vgat-Cas9 mice at the end of the in vivo series of studies. In keeping with the predominant GABAergic phenotype of preoptic area neurons (*Moffitt et al., 2018*), large numbers of cells expressing EGFP/Cas9 were detected throughout the region (*Figure 1A and B*), including the AVPV, PVpo, MPN, and bed nucleus of the stria terminalis. Analyses of the MPN in mice that did not receive AAV injections found that 52.1 ± 4.2% of VGAT-EGFP neurons expressed ESR1 (N=12).

The percentage of VGAT-EGFP neurons expressing ESR1 was evaluated within AAV-injected regions of the RP3V and MPN in mice receiving gRNA-2, gRNA-3, or control gRNA-LacZ. In keeping with prior studies showing that up to one half of medial preoptic area GABA neurons express ESR1 (*Herbison, 1997*; *Cheong et al., 2015*), we found in gRNA-LacZ mice that 54.9 ± 3.7% and 59.7 ± 3.9% of VGAT-EGFP neurons in the RP3V (N=6) and MPN (N=7), respectively, were immunoreactive for ESR1 (*Figure 1A and C*). In mice receiving gRNA-2, these values were reduced to 23.1 ± 3.3% (N=8) and 22.2 ± 4.7% (N=9) while gRNA-3 resulted in 17.6 ± 1.8% (N=8) and 19.3 ± 2.9% (N=8) of VGAT-EGFP neurons with ESR1 in the RP3V and MPN, respectively (*Figure 1B and C*). Inexplicably, one gRNA-3 mouse with correct AAV targeting maintained normal levels of ESR1 expression (52%) and was excluded from the analysis. Overall, gRNA-2 and -3 generated a significant 62% and 68% reduction ESR1 expression by RP3V VGAT neurons (p<0.0001; one-way ANOVA $F_{(2,19)}$ = 41.91, post

hoc Dunnett's test versus LacZ p<0.0001 [gRNA-2 and gRNA-3], *Figure 1C*) and a 63% and 71% reduction ESR1 expression by MPN VGAT neurons (p<0.0001; one-way ANOVA $F_{(2,21)}$ = 29.79, post hoc Dunnett's test versus LacZ p<0.0001 [gRNA-2 and gRNA-3], p<0.0001 [gRNA-3]; *Figure 1C*). Together, these data demonstrate that the two gRNAs result in a very similar 60–70% reduction in ESR1 expression in preoptic VGAT neurons.

## Locations of VGAT ESR1 knockdown

Bilateral AAV injections occurred in an heterogenous manner within the medial preoptic area of individual mice (*Figure 1D–G*). Due to the established importance of the RP3V in the estrogen-positive feedback mechanism and possible role of the MPN (*Wintermantel et al., 2006*; *Porteous and Herbison, 2019*), mice were categorized according to the unilateral or bilateral spread of AAV in the AVPV, PVpo, and MPN. Injection sites including parts of the AVPV and/or PVpo were termed an 'RP3V' hit while AAV spread in central aspects of the medial preoptic area involving at least part of the MPN and were termed an 'MPN' hit (see *Figure 1D–G*).

Nine gRNA-2 mice had injection sites involving both the RP3V and MPN; in three mice this included bilateral hits of the 'RP3V' and 'MPN' (*Figure 1D*) while four mice had one region with a bilateral hit and a unilateral or miss for the other brain region (*Figure 1E*). The remaining two mice received solely unilateral injections and were excluded from functional correlations. Eight gRNA-3 mice had injection sites involving RP3V and MPN. This included bilateral hits of the 'RP3V' and 'MPN' (*Figure 1F*) in three mice with another three mice having one region with a bilateral hit and the other unilateral. The remaining two gRNA-3 mice received solely unilateral injections and were excluded from functional correlations. Of eight gRNA-LacZ mice, four had bilateral injections involving the RP3V and/or MPN (*Figure 1G*). Two mice had unilateral injections involving both the AVPV and MPN but as ESR1 expression in VGAT neurons is not changed by gRNA-LacZ, injections were included in functional analyses. Two further gRNA-LacZ mice with small unilateral injections in only the MPN were excluded from

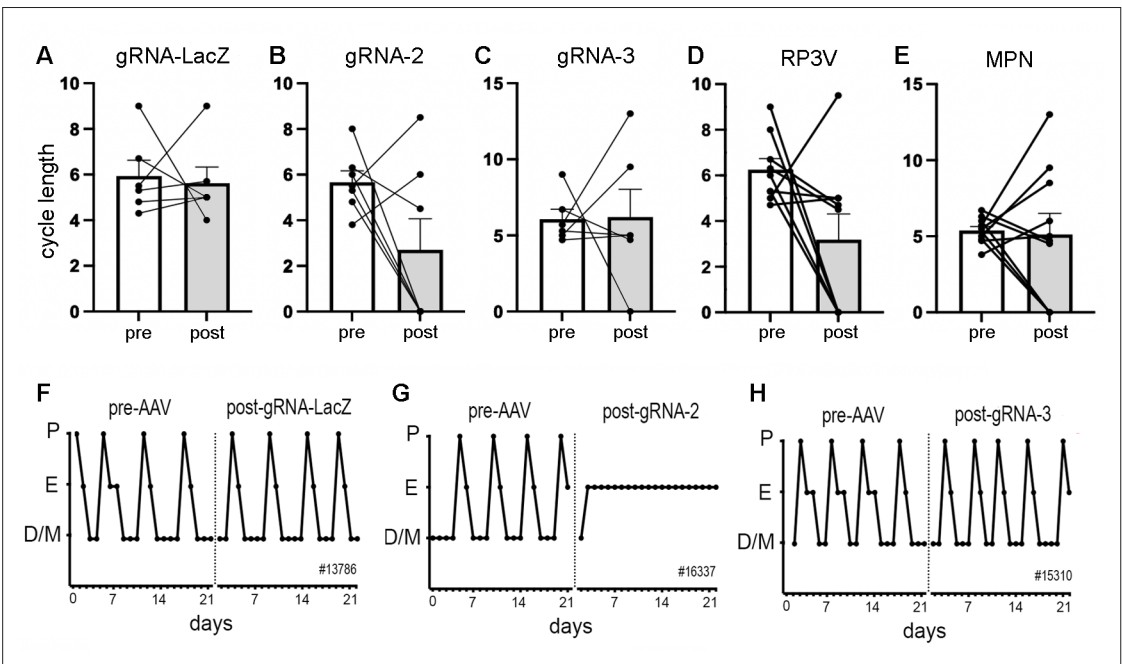

**Figure 2.** Deletion of ESR1 from preoptic GABA neurons and estrous cyclicity. (**A–C**) Individual paired data points (n=6-7) and mean ± SEM estrous cycle length before and after AAV gRNA injection of lacZ, gRNA-2, and gRNA-3 into the RP3V and medial preoptic nuclei (MPN). (**D–E**) Individual paired data points (n=9-10) and mean ± SEM estrous cycle length before and after AAV gRNA injection in mice with bilateral AAV injections in the RP3V and MPN analyzed separately. Four mice (3× gRNA-2, 1× gRNA-3) enter constant estrus and one gRNA-2 mouse was in constant diestrus, all scored as a cycle length of 0. No significant effects of gRNA injection were detected (p>0.05 Wilcoxon paired tests). (**F–H**), Examples of estrous cycle patterns from three mice including one (**G**) that entered constant estrous following gRNA-2 injection. The individual animal number is given in each frame. All data in *Source data 1*.

functional correlations. This resulted in experimental groups of six gRNA-LacZ, seven gRNA-2, and six gRNA-3 mice.

## Effects of VGAT ESR1 knockdown on estrous cyclicity

The estrous cycles of Vgat-Cas9 mice were determined over a 3-week period before stereotaxic injection of AAV-gRNA and then again for 3 weeks after a 3-week post-surgical interval. Mean cycle length for all animals prior to AAV injection was 5.9±1.5 days. No significant differences (p>0.05, Wilcoxon paired tests) were detected in cycle length for mice given gRNA-LacZ (n=6), gRNA-2 (N=7), or gRNA-3 (n=6) (*Figure 2A–C*). Whereas individual gRNA-LacZ mice exhibited relatively stable cycles before (5.9±0.7 days) and 3 weeks after (5.6±0.7 days) AAV injection (*Figure 2F*), gRNA-2 and -3 mice displayed considerable variability in estrous cycle length after AAV injection with some mice entering constant estrous or diestrus (*Figure 2G*); scored as zero for cycle length. We considered that this heterogeneity may have resulted from variations in ESR1 knockdown across the RP3V and MPN in individual mice. To assess this, all animals that had bilateral AAV injections involving the RP3V (n=9) and those with bilateral MPN injections (n=10) were considered as separate groups. These groupings also exhibited great variability with no significant differences (p>0.05, Wilcoxon paired tests) in cycle length (*Figure 2D and E*). The four mice displaying constant estrous did not exhibit any unique gRNA, bilateral/unilateral injection location, or level of ESR1 in RP3V or MPN VGAT neurons.

## Effects of VGAT ESR1 knockdown on pulsatile LH secretion

Following the post-AAV estrous cycle monitoring, pulsatile LH secretion was evaluated in diestrus, or in estrus for the four mice that had stopped cycling, using 6 min interval tail-tip bleeding for 180 min (*Steyn et al., 2013*; *Czieselsky et al., 2016*) and analyzed with PULSAR-Otago using the 'Intact Female' parameters (*Porteous et al., 2021*).

All mice exhibited typical profiles of pulsatile LH secretion (*Figure 3A–D*) with the exception of three mice (one in each gRNA group) that had no LH pulse during the 180 min sampling period. The gRNA-lacZ mice exhibited mean LH levels of 0.39±0.10 ng/mL (n=6) and had LH pulses with an interval of 32.6±3.1 min and amplitude of 0.92±0.17 ng/mL (n=5), as found in wild-type mice (*Czieselsky et al., 2016*). No differences were detected in mean LH (p=0.35, H=2.17 Kruskal-Wallis test), pulse amplitude (p=0.034, H=6.26 with no post hoc differences between gRNA-LacZ and gRNA-2 [p=0.883, Dunn's test] or gRNA-3 [p=0.207]), or pulse interval (p=0.34, H=2.24) between gRNA-LacZ and gRNA-2 (n=6–7) or gRNA-3 mice (n=5–6) (*Figure 3E–G*). To test whether bilateral ESR1 knockdown in VGAT neurons in the RP3V or MPN might be more selective in identifying a pulse phenotype, we re-grouped mice as above depending on their bilateral involvement of the RP3V and MPN and correlated pulse parameters with ESR1 expression in VGAT neurons. No significant correlations were detected between any parameter in either region (Pearson correlation r values of 0.06–0.41; p>0.05) (*Figure 3H–M*).

## Effects of VGAT ESR1 knockdown on the LH surge

After pulse bleeding studies, mice were ovariectomized (OVX) and given an estradiol replacement regimen that evokes the LH surge (*Czieselsky et al., 2016*) with a terminal blood sample analyzed for LH around the time of lights off and the brain processed for dual GnRH-cFos immunohistochemistry.

All gRNA-lacZ mice had increased cFos expression in rostral preoptic area GnRH neurons (43 ± 8%, range 24–84%; 21±0.7 GnRH neurons/section), although two of the mice had LH levels below 1.5 ng/mL indicating that they had likely surged before lights off (*Figure 4A and B*). Unexpectedly, huge variation was found between gRNA-2 and gRNA-3 treated mice in the generation of the surge (*Figure 4A, B*). The number of GnRH neurons detected in the rPOA was the same in all groups (gRNA-2, 20±1.9 GnRH neurons/section; gRNA-3, 20±1.7 GnRH neurons/section). However, a significant reduction in the number GnRH neurons with cFos was found (p=0.0005, H=11.81 Kruskal-Wallis) with gRNA-2 (p=0.0023 Dunn's multiple comparisons) but not gRNA-3 (p=0.91) compared with gRNA-LacZ mice (*Figure 4A*). The wide variation in single-point LH values in all three groups did not reveal any significant differences between groups (*Figure 4B*). This was a curious result given that gRNA-2 and -3 generated an identical 60–70% ESR1 knockdown in VGAT neurons (*Figure 1C*). To explore this effect of gRNA-2 further and parcellate between the RP3V and MPN, we again re-grouped the mice depending on whether they had bilateral involvement of the RP3V and MPN. A significant correlation

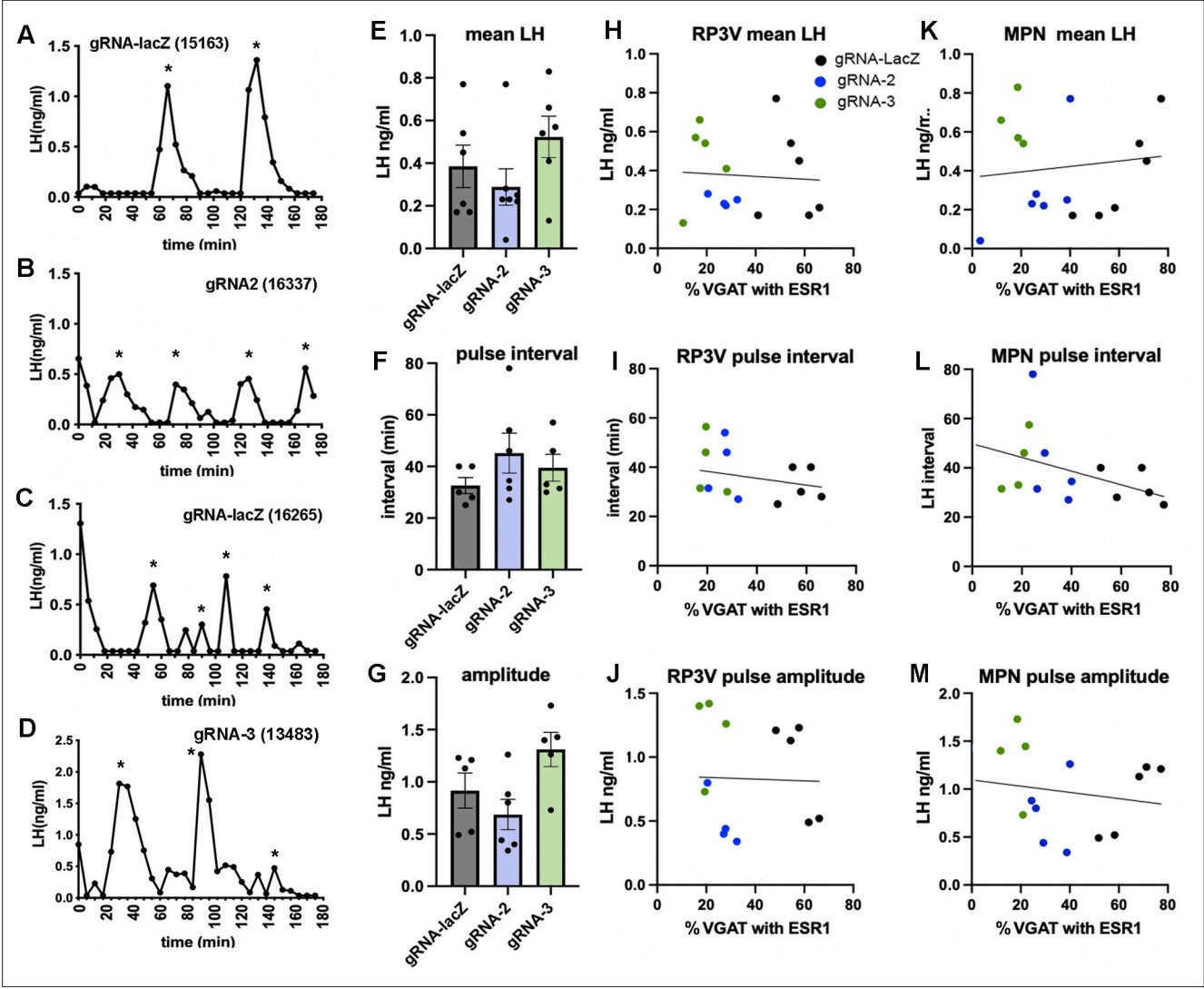

**Figure 3.** Deletion of ESR1 from preoptic GABA neurons does not alter pulsatile luteinizing hormone (LH) secretion. (**A–D**) Representative LH pulse profiles from female mice given AAV gRNA-lacZ, gRNA-2, and gRNA-3. The mouse identification number is given in brackets. (**E–G**) Histograms show the individual data points (n=5-6) and mean ± SEM for parameters of pulsatile LH secretion in mice given gRNA-lacZ, gRNA-2, and gRNA-3 into the RP3V and medial preoptic nuclei (MPN). No significant effects are detected (p>0.05, Kruskal-Wallis test). (**H–J**) Correlations between the % VGAT neurons with ESR1 in the RP3V and parameters of pulsatile LH secretion. Individual mice are color-coded according to their gRNA treatment. No significant correlations were detected (Pearson r<0.34 in all cases). (**K–M**) Correlations between the % VGAT neurons with ESR1 in the MPN and parameters of pulsatile LH secretion. Individual mice are color-coded according to their gRNA treatment. No significant correlations were detected (Pearson r<0.41 in all cases). All data in *Source data 1*.

was found between ESR1 expression in RP3V VGAT neurons and cFos in GnRH neurons (p=0.008, Pearson r=0.66) (*Figure 4C*) but not LH secretion (p=0.26, Pearson r=0.31) across all mice (*Figure 4D*). No significant correlation was found for ESR1 expression in MPN VGAT neurons and cFos in GnRH (p=0.18, Pearson r=0.36) or LH secretion (p=0.60, Pearson r=0.14) (*Figure 4E and F*).

## CRISPR knockdown of ESR1 in RP3V GABA-kisspeptin neurons

The above observations suggested that ESR1 in RP3V VGAT neurons was required for the activation of GnRH neurons at the time of the surge. However, it remained unclear why so much heterogeneity remained in this group; some mice with significant ESR1 knockdown (>80%) continued to show surge levels of cFos in GnRH neurons (>40%). Further, the mice entering constant estrus had some of the highest levels of remaining ESR1 in RP3V VGAT neurons. Given the importance of RP3V kiss-peptin neurons for the GnRH surge (*Clarkson et al., 2008*; *Wang et al., 2019*) and evidence that

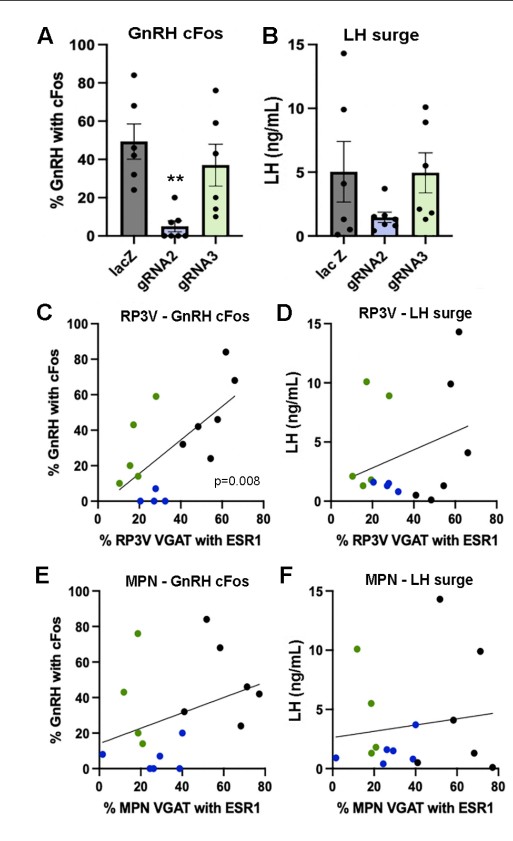

**Figure 4.** Effects of ESR1 deletion in preoptic GABA neurons on surge parameters. (**A, B**) Individual data points (n=6-7) and mean ± SEM values showing the percentage of gonadotropin-releasing hormone (GnRH) neurons with cFos and single-point luteinizing hormone (LH) levels for mice killed at the time of the expected surge given gRNA-LacZ (black), gRNA-2 (blue), and gRNA-3 (green) injections centered on the RP3V and medial preoptic nuclei (MPN). **p=0.0023 (Krusakl-Wallis) compared with LacZ. (**C, D**) Correlations between the % of RP3V VGAT neurons with ESR1 and cFos expression by GnRH neurons or LH secretion. Individual mice are color-coded according to their gRNA treatment. A significant correlation for cFos in GnRH neurons exists (p=0.008, Pearson r=0.66) but not for LH (p=0.26, Pearson r=0.31). (**E, F**) Correlations between the % of MPN VGAT neurons with ESR1 and cFos expression by GnRH neurons or LH secretion. Individual mice are color-coded according to their gRNA treatment. No significant correlations were found. All data in **Source data 1**.

these cells are a GABAergic phenotype (**Moffitt et al., 2018**; **Stephens and Kauffman, 2021**), we considered that one possible explanation for the heterogeneity was that the CRISPR gene editing had knocked down ESR1 variably in the sub-population of RP3V GABA neurons co-expressing kisspeptin.

## ESR1 knockdown in preoptic kisspeptin neurons

The last set of brain sections from all gRNA mice underwent dual-label immunohistochemistry to assess ESR1 expression in RP3V kisspeptin neurons. The normal periventricular distribution of kisspeptin neurons was detected within the AVPV and PVpo of gRNA-lacZ mice (N=6) with 20.1±2.0 and 24.0±1.9 kisspeptin cells/section, respectively, and 74.5 ± 3.3% and 63.3 ± 5.1% of kisspeptin neurons positive for ESR1 immunoreactivity ('lacZ' group, **Table 1**; **Figure 5**). The 17 mice receiving gRNA-2 or gRNA-3 clearly fell into one of three groups:

• 'Normal' group (2× gRNA-2, 5× gRNA-3) exhibited a normal number of kisspeptin neurons in the AVPV and PVpo with normal ESR1 expression (**Table 1**).
• 'Unilateral' group (4× gRNA-2, 2× gRNA-3) had a substantial >75% reduction in kisspeptin-immunoreactive neuron number on only one side of the brain in either the AVPV (p=0.007, Kruskal-Wallis ANOVA) and/or the PVpo (p=0.002, Kruskal-Wallis with Dunn's tests) (**Table 1**). The expression of ESR1 in remaining kisspeptin neurons was highly variable (**Table 1**).
• 'Bilateral' group (3× gRNA-2, 1× gRNA-3) had almost no kisspeptin neurons apparent in either the AVPV (0.4±0.1 kisspeptin neurons/section; p=0.0006) or PVpo (0.8±0.3 kisspeptin neurons/section; p=0.003) (**Table 1**; **Figure 5**).

We also determined ESR1 knockdown in RP3V and MPN VGAT neurons in these new groups.

As expected, the 'lacZ' mice were normal with ESR1 expressed in 50–61% of VGAT neurons while both the 'normal' and 'unilateral' groups had 18% co-expression (p=0.0002–0.0242, Kruskal-Wallis ANOVA), and the 'bilateral' group exhibited an average of 25% co-expression that was not different to either 'normal' or 'unilateral' groups (**Table 2**).

## Effects of ESR1 knockdown on estrous cycles, pulsatile LH secretion, and the LH surge

The groups segregated by kisspeptin expression exhibited highly consistent reproductive phenotypes. The two gRNA groups with normal RP3V kisspeptin expression ('lacZ' and 'normal') exhibited normal estrous cycles 3 weeks after AAV injection (**Figure 5**), the usual variable LH surge levels

**Table 1.** Kisspeptin-ESR1 co-expression in re-grouped gRNA mice.

Table showing the numbers of kisspeptin neurons/section in the anteroventral periventricular (AVPV) and preoptic periventricular (PVpo) and percentage expression with ESR1. 'LacZ' refers to all mice given gRNA-LacZ, 'normal' refers to all gRNA-2/3 mice with normal kisspeptin expression (unilateral cell counts shown), 'unilateral' refers to gRNA-2/3 mice in which only one side of the brain had reduced kisspeptin cell numbers with the cell count on the affected side given, 'bilateral' refers to gRNA-2/3 mice with essentially no cytoplasmic kisspeptin expression bilaterally in the RP3V. Too few kisspeptin neurons existed to reliably determine co-expression with ESR1. **p<0.01, ***p<0.001 compared with lacZ group (Kruskal-Wallis with Dunn's tests, exact p-values given below). All data in *Source data 1*.

|  |  | LacZ (n=8) | Normal (n=7) | Unilateral (n=6) | Bilateral (n=4) |
|---|---|---|---|---|---|
| AVPV | No. kisspeptin neurons/section | 20.1±2.0 | 15.4±0.9 | 5.6±2.5** (p=0.0066) | 0.4±0.1*** (p=0.0006) |
|  | % Kiss with ESR1 | 74.5±3.3 | 78.5±4.4 | 57.3±19.7 | – |
| PVpo | No. kisspeptin neurons/section | 24.0±1.9 | 19.6±1.3 | 4.8±2.8** (p=0.0016) | 0.8±0.3** (p=0.0031) |
|  | % Kiss with ESR1 | 63.3±5.1 | 70.5±5.2 | 27.5±17.1 | – |

(4.8±1.7 and 6.3±1.3 ng/mL; range 0.1–14.3 ng/mL) (*Figure 5*) and ~40–50% of GnRH neurons with cFos at the time of the expected LH surge (*Figure 5*). This was accompanied by normal pulsatile LH (*Figure 5*). In contrast, the 'bilateral' mice with essentially no RP3V kisspeptin expression were all found to enter constant estrus (*Figures 2 and 5*) and have both low LH levels (1.4±0.3 ng/mL;

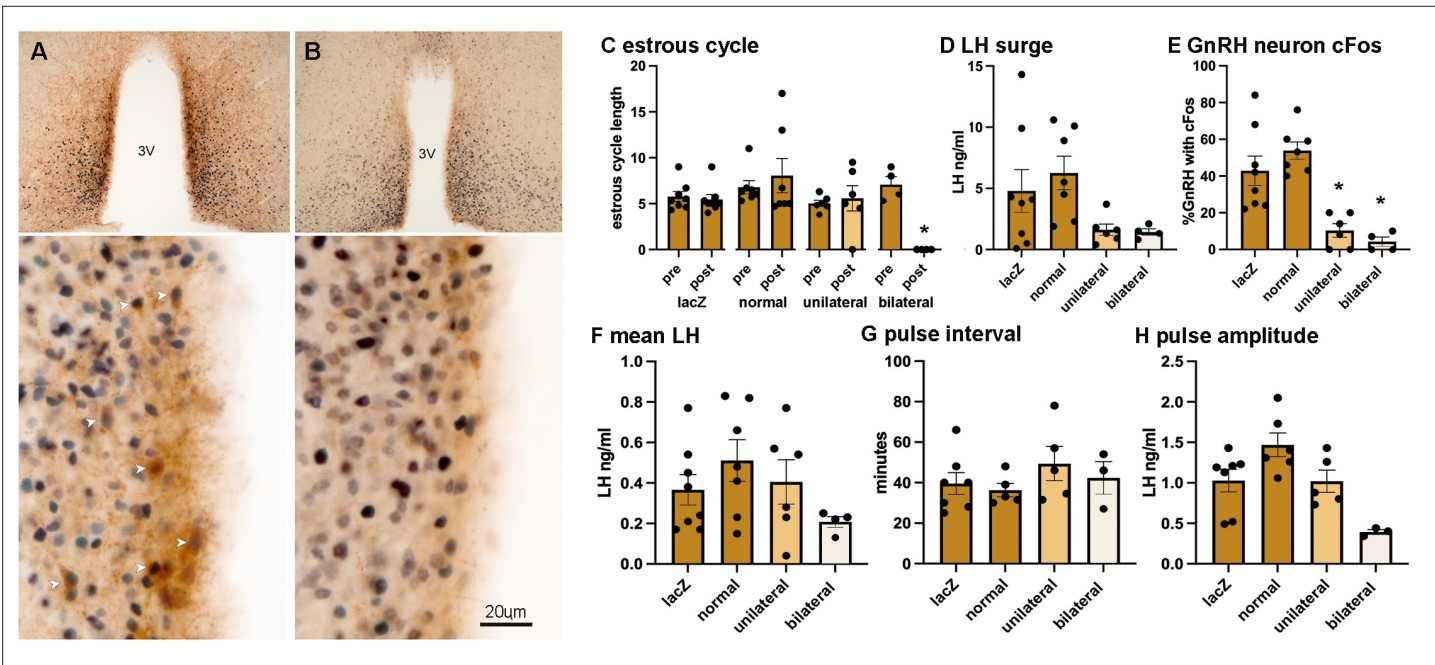

**Figure 5.** Suppression of RP3V kisspeptin expression is associated with the loss of estrous cycles and the surge mechanism. (**A, B**) Dual-label immunohistochemistry for kisspeptin (brown) and ESR1 (black) shows the normal high level of co-expression (white arrowheads) in a representative gRNA-lacZ mouse (**A**) but near absence of kisspeptin immunoreactivity in a representative 'bilateral loss' mouse (#16337) (**B**). 3V, third ventricle. Scale bar in B is the same for A. (**C**) Individual data points (n=4-7) and mean ± SEM values showing estrous cycle length before and after gRNA injection in gRNA 'lacZ' mice, gRNA-2/3 mice with 'normal' kisspeptin expression, gRNA-2/3 mice with a 'unilateral' reduction in kisspeptin, and gRNA-2/3 mice with a near-complete 'bilateral' loss of kisspeptin. *p<0.05 compared to pre-values. (**D, E**) Individual data points (n=4-7) and mean ± SEM single-point luteinizing hormone (LH) levels and % of gonadotropin-releasing hormone (GnRH) neurons with cFos at the time of the expected surge in gRNA 'lacZ' mice, gRNA-2/3 mice with 'normal' kisspeptin expression, gRNA-2/3 mice with a 'unilateral' reduction in kisspeptin, and gRNA-2/3 mice with a near-complete 'bilateral' loss of kisspeptin. *p<0.05 compared to lacZ. (**F–H**) Individual data points (n=3-7) and mean ± SEM parameters of pulsatile LH secretion in gRNA 'lacZ' mice, gRNA-2/3 mice with 'normal' kisspeptin expression, gRNA-2/3 mice with a 'unilateral' reduction in kisspeptin, and gRNA-2/3 mice with a near-complete 'bilateral' loss of kisspeptin. No significant differences were detected. All data in *Source data 1*.

**Table 2.** VGAT-ESR1 in mice grouped on the basis of kisspeptin expression.
Table showing the percentage of VGAT neurons in the RP3V and MPN expressing ESR1 in gRNA mice re-grouped on the basis of kisspeptin expression (see *Table 1* for explanation of groups). *p<0.05, ***p<0.001 compared with lacZ group (Kruskal-Wallis with Dunn's tests, exact p-values given below). All data in *Source data 1*.

|  |  | LacZ (n=8) | Normal (n=7) | Unilateral (n=6) | Bilateral (n=4) |
|---|---|---|---|---|---|
| RP3V | % VGAT with ESR1 | 50.4±6.1 | 17.9±2.3* (p=0.0242) | 17.7±3.0* (p=0.0358) | 24.5±4.8 |
| MPN | % VGAT with ESR1 | 61.2±4.2 | 14.0±3.1*** (p=0.0002) | 22.0±5.1* (p=0.0162) | 30.1±4.8 |

range 0.8–2.1 ng/mL) and low cFos expression in GnRH neurons (4.3 ± 2.6%; p=0.0004, H=18.24, Kruskal-Wallis with p=0.0185 post hoc Dunn's test) (*Figure 5*) at the time of the expected surge. The LH surge levels across the groups were significantly different (p=0.0320, H=8.802, Kruskal-Wallis ANOVA; *Figure 5*) but the high variability in lacZ and normal mice prevented significant differences between individual groups (p=0.698, post hoc Dunn's test). Pulsatile LH release in 'bilateral' mice was variable with one mouse having no pulses in the 180 min recording period while the others displayed pulses not significantly different to gRNA-lacZ mice despite an apparent trend for reduced LH pulse amplitude (p=0.1409) (*Figure 5*). Finally, the 'unilateral' mice displayed a phenotype intermediate between that of 'normal' and 'bilateral' groups. Estrous cyclicity was normal (*Figure 5*) except for one mouse that went into constant diestrus and LH pulsatility was unaffected (*Figure 5*), while LH levels (1.6±0.5 ng/mL) and the % GnRH neurons with cFos (10.3 ± 3.7%, p=0.0297, post hoc Dunn's tests) at the time of the expected LH surge were similar to the 'bilateral' group.

## Discussion

We find here that CRISPR knockdown of ESR1 in preoptic area GABAergic neurons results in a variable reproductive phenotype with most mice exhibiting normal estrous cycles and LH secretion. However, re-sorting individual mice on the basis of their RP3V kisspeptin expression provided highly consistent reproductive traits; mice with absent RP3V kisspeptin expression were acyclic and failed to exhibit an LH surge but retained pulsatile LH secretion. These data demonstrate an essential role for the RP3V GABA-kisspeptin neuronal phenotype in the murine estrogen-positive feedback mechanism.

We show here that gRNA-2 and gRNA-3 are effective at knocking down ESR1 expression in preoptic GABA neurons in adult female mice; both gRNA achieved a 60–70% reduction in ESR1 within the RP3V and MPN compared to gRNA-lacZ mice. We had previously reported that gRNA-3 generated an 80% knockdown in ESR1 within arcuate kisspeptin neurons (*McQuillan et al., 2022*). This indicates the general utility of the CRISPR approach for in vivo gene editing of ESR1 in the mouse brain.

When examined on the basis of the original GABA neuron groupings, the only significant observation was that expression of ESR1 in RP3V GABA neurons correlated positively with the percentage of GnRH neurons expressing cFos at the time of the surge. The significant correlation between the ESR1 expression in RP3V GABA neurons and cFos in GnRH neurons when all gRNA-2 and gRNA-3 mice were combined revealed that this was not likely to be due to any unique actions of gRNA-2. While providing evidence that ESR1 in RP3V GABA neurons was required for normal activation of the GnRH neurons at the time of the surge, this was not the case for the LH surge. We note that single-point LH measurements are unreliable even for the OVX+E2 paradigm with individual mice exhibiting quite marked variability in surge onset and peak LH levels (*Czieselsky et al., 2016*). As such, we favor cFos expression in GnRH neurons as a more reliable index of whether the surge has commenced at the time of investigation.

The highly variable reproductive phenotypes of the ESR1 knockdown mice became consistent when sorted on the basis of their RP3V kisspeptin expression. Four mice exhibited a near-complete lack of kisspeptin immunoreactivity within the entire RP3V. As the RP3V kisspeptin neurons are a GABAergic phenotype (*Moffitt et al., 2018*), they will all be targeted for Cas9 expression in Vgat-Cas9 mice. Prior estimates have indicated that 20–75% of RP3V kisspeptin neurons express *Vgat* in adult mice (*Cravo et al., 2011*; *Cheong et al., 2015*) although it is very likely that all RP3V kisspeptin neurons

express VGAT during development and, as such, will all express Cas9 as adults. Hence, when the bilateral stereotaxic AAV-gRNA injections happen to include the entire extent of the RP3V, all GABA-kisspeptin neurons will be targeted. The almost complete lack of kisspeptin immunoreactivity is due to the critical dependence of kisspeptin expression on ESR1 in RP3V kisspeptin neurons (*Smith et al., 2005*; *Dubois et al., 2015*; *Greenwald-Yarnell et al., 2016*). The CRISPR-mediated knockdown in ESR1 within these RP3V cells would gradually result in the suppression of kisspeptin biosynthesis. However, one curious observation is that essentially all cytoplasmic kisspeptin immunoreactivity is absent despite the expectation that some kisspeptin neurons would retain functional ESR1 due to the inherent mosaicism of this CRISPR strategy (*Platt et al., 2014*). Hence, the near-complete absence of kisspeptin might also result from disordered circulating estradiol levels in acyclic mice.

We also noted an interesting reproductive phenotype in mice with unilateral knockdown of ESR1 in RP3V kisspeptin neurons resulting in unilateral reductions in kisspeptin-immunoreactive neuron numbers. Although showing no change in estrous cycles, these mice had somewhat similar deficits in GnRH neuron activation levels at the time of the surge compared to mice with no RP3V kisspeptin. Prior studies have indicated that 30–50% reductions in RP3V kisspeptin neuron number are compatible with a normal surge mechanism (*Szymanski and Bakker, 2012*; *Hu et al., 2015*). It is possible that a substantial reduction in RP3V kisspeptin input to GnRH neurons from one side of the brain results in weaker surge activation. It may also alter the timing of surge onset outside the window examined in this study as the mice maintain normal estrous cycles implying that ovulation is occurring.

It is widely anticipated that that RP3V kisspeptin neurons are the critical estradiol-sensing component underlying the preovulatory surge mechanism (*Uenoyama et al., 2021*; *Goodman et al., 2022*; *Kauffman, 2022*). We now provide data demonstrating that ESR1-expressing RP3V kisspeptin neurons are essential for the surge mechanism and estrous cyclicity in mice. It is unclear why the previous CRISPR-Cas9 study by Wang and colleagues did not observe the same result (*Wang et al., 2019*). However, that study did not examine kisspeptin expression, so it is difficult to estimate the degree or level of functional knockout within the population. Also, only the AVPV was targeted so that the great majority of RP3V kisspeptin neurons that are located in the PVpo (*Clarkson and Herbison, 2006*) were presumably unaffected in that study.

We find that pulsatile LH secretion is maintained in the absence of RP3V kisspeptin expression. A non-significant trend toward reduced LH pulse amplitude was observed but pulse frequency was unchanged. This observation agrees with the 'two-compartment model' of the GnRH neuron network in which the kisspeptin pulse and kisspeptin surge generators are thought to operate in relative independence at different compartments of the female GnRH neuron to bring about LH pulses and the LH surge, respectively (*Herbison, 2020*).

A caveat to an exclusive role of RP3V kisspeptin neurons in surge generation is the possibility that non-kisspeptin ESR1-expressing GABA neurons immediately adjacent to RP3V kisspeptin cells are also required for the surge. However, we note that acyclic mice with no kisspeptin ('bilateral loss') had similar levels of ESR1 knockdown in RP3V GABA neurons found in normal mice that exhibited normal surge activation. Nevertheless, we cannot be sure that the significant correlation between the degree of ESR1 knockdown in RP3V GABA neurons and GnRH neuron activation at the surge is attributable entirely to GABA-kisspeptin neurons. If RP3V GABAergic non-kisspeptin neurons are involved, it seems that they may play an intrinsic role within the RP3V as GABA release from the RP3V to GnRH neurons is inconsequential for surge induction (*Piet et al., 2018*).

In summary, we show here that CRISPR gene editing can be used to efficiently knock down ESR1 in GABAergic neurons. Characterizing reproductive phenotypes on an individual animal basis in relation to GABA neuron sub-populations revealed that a loss of kisspeptin in RP3V GABA neurons was perfectly correlated with an absent surge mechanism and estrous acyclicity. These observations provide definitive evidence for ESR1-expressing RP3V GABA-kisspeptin neurons being essential for the preovulatory surge mechanism while having no role in pulse generation.

## Methods
### Animals
*Slc32a1^Cre^, Rosa26^LSL-Cas9-EGFP^* (Vgat-Cas9) mice were generated by crossing 129S6Sv/Ev C57BL6 *Slc32a1^Cre^* mice (JAX stock #026175) (*Vong et al., 2011*) with B6J.129(B6N) *Rosa26^LSL-Cas9-EGFP^* mice

(JAX stock #026175) (*Platt et al., 2014*). All mice were provided with environmental enrichment under conditions of controlled temperature (22 ± 2°C) and lighting (12 hr light/12 hr dark cycle; lights on at 6:00 hr and off at 18:00 hr) with ad libitum access to food (Teklad Global 18% Protein Rodent Diet 2918, Envigo, Huntingdon, UK) and water. Daily vaginal cytology was used to monitor the estrous cycle stage. All animal experimental protocols were approved by the Animal Welfare Committee of the University of Otago, New Zealand (96/2017).

## Experimental protocol

The estrous cycles of adult female Vgat-Cas9 mice were assessed for 3 weeks and mice exhibiting regular 4- to 6-day cycles given bilateral stereotaxic injections of AAV1-U6-gRNA-LacZ/ESR1-2/ESR1-3-Ef1α-mCherry into the medial preoptic area. The gRNA sequences and generation of AAVs have been detailed previously (*McQuillan et al., 2022*). Three weeks later, estrous cycles were again monitored for 3 weeks. Pulsatile LH secretion was then assessed using 6 min tail-tip bleeding for 180 min. Mice were then ovariectomized and given an estrogen replacement regimen to induce the LH surge at which time they were killed, a terminal blood sample taken, and perfusion-fixed for immunohistochemical analysis.

## Stereotaxic surgery and injections

Adult mice (3–4 months of age) were anesthetized with 2% isoflurane, given local lidocaine (4 mg/kg, s.c.) and carprofen (5 mg/kg, s.c.) and placed in a stereotaxic apparatus. Bilateral injections of 1.5 µL AAV1-U6-gRNA-LacZ/ESR1-2/ESR1-3-Ef1α-mCherry-WPRE-SV40 (1.3–2.5 × 10$^{13}$ GC/mL) were given into the medial preoptic area and mice allowed to recover for 3 weeks before commencing the experimental protocol. A custom-made bilateral Hamilton syringe apparatus holding two 29-gauge needles 0.9 mm apart was used to perform bilateral injections into the preoptic area. The needles were lowered into place over 2 min and left in situ for 3 min before the injection was made. The AAV was injected at a rate of ~100 nL/min with the needles left in situ for 10 min before being withdrawn. Carprofen (5 mg/kg body weight, s.c.) was administered for post-operative pain relief for 2 days.

## Pulsatile hormone measurement, ovariectomy, estrogen replacement, and LH assay

Profiles of pulsatile LH secretion were determined by tail-tip bleeding at 6 min intervals for 180 min in diestrus, or where cycles had stopped, estrous-stage mice as reported previously (*Czieselsky et al., 2016*). Bilateral ovariectomy was performed under isoflurane anesthesia with pre- and post-operative carprofen (5 mg/kg body weight, s.c.). Estradiol replacement was provided by s.c. implantation of an ~1 cm length of silastic capsule (Dow Corning, USA) filled with 0.4 µg/mL 17-β-estradiol to provide 4 µg 17-β-estradiol/20 g body weight. This protocol returns the plasma profile of pulsatile LH secretion and 17-β-estradiol concentrations to that found in diestrus females (*Porteous et al., 2021*). Six days later, mice were given an s.c. injection of estradiol benzoate (1 µg in 100 µL) at 09:00 and killed at 19:00 the following evening at the time of lights off (*Czieselsky et al., 2016*) when a terminal blood sample was taken. Plasma LH concentrations were determined using an ultrasensitive LH ELISA (*Steyn et al., 2013*; *Czieselsky et al., 2016*) and had an assay sensitivity of 0.04 ng/mL and intra- and inter-assay coefficients of variation of 4.6% and 9.3%. Pulse analysis was undertaken with PULSAR Otago (*Porteous et al., 2021*) using the following validated parameters for intact female mice: smoothing 0.7, peak split 2.5, level of detection 0.04, amplitude distance 3 or 4, assay variability 0, 2.5, and 3.3, G values of 3.5, 2.6, 1.9, 1.5, and 1.2.

## Immunohistochemistry

Mice were given a lethal overdose of pentobarbital (3 mg/100 µL, i.p.) and perfused through the heart with 20 mL of 4% paraformaldehyde in phosphate-buffered saline. Three sets of 30-µm-thick coronal sections were cut through the full extent of the preoptic area and incubated overnight in a cocktail of chicken anti-EGFP (1:5000; AB13970, Abcam; RRID:AB_300798) and rabbit anti-ESR1 (1:1000; #06-935, Merck Millipore; RRID:AB_310305) followed by 2 hr incubation in goat anti-chicken 488 (1:400; Molecular Probes) and biotinylated goat anti-rabbit immunoglobulins (1:400, Vector Laboratories) and then 2 hr in Streptavidin 647 (1:400, Molecular Probes). This was followed by placing in 5% normal goat serum for 1 hr and labeling for mCherry with rabbit anti-mCherry

(1:10,000; Abcam; RRID:AB_2571870) and goat anti-rabbit 568 (1:400; Molecular Probes) using the same incubation periods. The use of a second anti-rabbit antisera resulted in cytoplasmic 568/mCherry labeling that was used for defining the extent of AAV transduction and potentially some 568 fluorescence in ESR1-positive nuclei that was avoided by analyzing only in the far-red 647 nm spectrum.

To assess ESR1 expression in kisspeptin neurons, dual-label chromogen immunohistochemistry was undertaken with rabbit anti-ESR1 (1:5000), and peroxidase-labeled goat anti-rabbit (1:400, Vector Labs) revealed with nickel-DAB followed by rabbit anti-kisspeptin antisera (AC566, 1:10,000; RRID:AB_2314709), and biotinylated goat anti-rabbit immunoglobulins (1:400, Vector Labs) and Vector Elite avidin-peroxidase (1:100) with DAB as the chromogen.

To assess activation of GnRH neurons, dual-label chromogen immunohistochemistry was undertaken with rabbit anti-cFos (sc-52, 1:5000, Santa Cruz; RRID:AB_2106783) and biotinylated goat anti-rabbit immunoglobulins (1:400, Vector Labs) and Vector Elite avidin-peroxidase (1:100) with NiDAB, followed by rabbit anti-GnRH (GA01, 1:1000, RRID:AB_2721114) and peroxidase-labeled goat anti-rabbit (1:400, Vector Labs) revealed with DAB.

Quantitative analyses of ESR1 expression in the RP3V and MPN neurons were undertaken on confocal images captured on a Nikon A1R multi-photon laser scanning microscope using 40× Plan Fluor, N.A. 0.75 objective using software Nikon Elements C. The numbers of EGFP-labeled cells with and without immunoreactive ESR1 nuclei were counted by an investigator blind to the experimental groupings. Cell counts were undertaken by analyzing all EGFP-positive cells through 10 z-slices of 2 μm thickness in sections at each of the levels of the RP3V-MPN for each mouse. The number of kisspeptin neurons with ESR1 was assessed under brightfield microscopy by counting the number of kisspeptin-immunoreactive cells (brown DAB) with and without black (nickel-DAB) ESR1-positive nuclei bilaterally in three to four sections throughout the AVPV and PVpo in each mouse. The number of GnRH neurons with cFos was assessed under brightfield microscopy by counting the number of GnRH-immunoreactive cells (brown DAB) with and without black (NiDAB) cFos-positive nuclei bilaterally in two sections of the rostral preoptic area in each mouse.

## Statistical analysis

Sample size was determined from prior studies without power analyses. All mice were randomized to experimental groups and experimenters were blind to groupings. Seven individual cohorts of experimental mice (n=3–7 in each) included gRNA ESR1-2 and gRNA LacZ mice, or gRNA ESR1-3 and gRNA LacZ mice. Excluded mice are justified in the appropriate Results section. Statistical analysis was undertaken on Prism 10 using parametric or non-parametric tests as appropriate. This included one-way ANOVA with post hoc Dunnett's test, Kruskal-Wallis with post hoc Dunn's tests, Wilcoxon paired tests, and Pearson correlations as appropriate depending on the normality of distribution. Data are presented as mean ± SEM.

## Acknowledgements

This work was supported by the New Zealand Marsden Fund (16-UOO-201), New Zealand Health Research Council (17-285), and the Wellcome Trust (10.35802/212242).

## Additional information

### Funding

| Funder | Grant reference number | Author |
| --- | --- | --- |
| Marsden Fund | 16-UOO-201 | Allan E Herbison |
| Health Research Council of New Zealand | 17-285 | Allan E Herbison |
| Wellcome Trust | 10.35802/212242 | Allan E Herbison |

| Funder | Grant reference number | Author |
| --- | --- | --- |

The funders had no role in study design, data collection and interpretation, or the decision to submit the work for publication. For the purpose of Open Access, the authors have applied a CC BY public copyright license to any Author Accepted Manuscript version arising from this submission.

## Author contributions

Jenny Clarkson, Data curation, Formal analysis, Investigation, Methodology, Writing – review and editing; Siew Hoong Yip, Formal analysis, Investigation, Writing – review and editing; Robert Porteous, Formal analysis, Investigation; Alexia Kauff, Resources, Methodology; Alison K Heather, Resources, Methodology, Writing – review and editing; Allan E Herbison, Conceptualization, Formal analysis, Supervision, Funding acquisition, Writing – original draft, Writing – review and editing

## Author ORCIDs

Allan E Herbison ⬤ https://orcid.org/0000-0002-9615-3022

## Ethics

This study was performed in strict accordance with experimental protocols approved by the Animal Welfare Committee at the University of Otago, NZ under license 96/2017. All surgery was performed under Isoflurane anesthesia, and every effort was made to minimize suffering.

Reviewer #1 (Public Review): https://doi.org/10.7554/eLife.90959.3.sa1
Reviewer #2 (Public Review): https://doi.org/10.7554/eLife.90959.3.sa2
Reviewer #3 (Public Review): https://doi.org/10.7554/eLife.90959.3.sa3
Author Response https://doi.org/10.7554/eLife.90959.3.sa4

# Additional files

## Supplementary files

- Source data 1. All measured parameters for each individual mouse.

- MDAR checklist

## Data availability

All data generated or analysed during this study are included in the manuscript and supporting file; *Source data 1* has been provided.

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
