## [Editor Report · eLife assessment]

This **important** study provides **convincing** evidence of the criticality of estradiol - estrogen receptor-mediated upregulation of kisspeptin within neurons of the preoptic area to generate an ovulation-inducing luteinizing hormone surge. The use of in vivo CRIPSR-Cas9 is novel in this system and provides a road map for future studies in reproductive neuroendocrinology. This paper will be of interest to reproductive neuroscientists and endocrinologists.

---

## [Referee Report · Reviewer #1 (Public Review)]

Summary: The current study examines the necessity of estrogen receptor alpha (ESR1) in GABA neurons located in the anteroventral and preoptic periventricular nuclei and the medial preoptic nucleus of hypothalamus. This brain area is implicated in regulating the pre-ovulatory LH surge in females, but the identity of the estrogen-sensitive neurons that are required remains unknown. The data indicate that approximately 70% knockdown of ESR1 in GABA neurons resulted in variable reproductive phenotypes. However, when the ESR1 knockdown also results in a decrease in kisspeptin expression by these cells, the females had disrupted LH surges, but no alterations in pulsatile LH release. These data support the hypothesis that kisspeptin cells in this region are critical for the pre-ovulatory LH surge in females.

Strengths: The current study examined the efficacy of two guide RNAs to knockdown ESR1 in GABA neurons, resulting in an approximate 70% reduction in ESR1 in GABA neurons. The efficacy of this knockdown was confirmed in the brain via immunohistochemistry and the reproductive outcomes were analyzed several ways to account for differences in guide RNAs or the precise brain region with the ESR1 knockdown. The analysis was taken one step further by grouping mice based on kisspeptin expression following ESR1 knockdown and examining the reproductive phenotypes. Overall, the aims of the study were achieved, the methods were appropriate, and the data were analyzed extensively. This data supports the hypothesis that kisspeptin neurons in the anterior hypothalamus are critical for the preovulatory LH surge.

Weaknesses: One minor weakness in this study is the conclusion that the two different guide RNAs didn't seem to have unique effects on GnRH cFos expression or the reproductive phenotypes. Though the data indicate a 60-70% knockdown for both gRNA2 and gRNA3, 3 of the 4 gRNA2 mice had no cFos expression in GnRH neurons during the time of the LH surge, whereas all mice receiving gRNA3 had at least some cFos/GnRH co-expression. In addition, when mice were re-categorized based on reduction (>75%) in kisspeptin expression, most of the mice in the unilateral or bilateral groups received gRNA2, whereas many of the mice that received gRNA3 were in the "normal" group with no disruption in kisspeptin expression. Whether these results occurred by chance or due to differences in the gRNAs remains unknown. Thus, additional experiments with increased sample sizes would be needed, even if the efficacy of the ESR1 knockdown was comparable, before concluding these 2 gRNAs don't have unique actions.

---

## [Referee Report · Reviewer #2 (Public Review)]

Clarkson et al investigated the impact of in vivo ESR1 gene disruption selectively in preoptic area GABA neurons on the estrogen regulation of LH secretion. The hypothalamic pathways by which estradiol controls the secretion of gonadotrophins are incompletely understood and relevant to a better understanding of the mechanisms driving fertility and reproduction. Using CRISPR-Cas9 methodology, the authors were able to effectively reduce the expression of estrogen receptor (ER)-alpha in GABA neurons located in the preoptic area of adult female mice. The results obtained were rather variable except in the animals with concomitant suppression of kisspeptin in the rostral periventricular region of the third ventricle (RP3V), which displayed interruption of ovarian cyclicity and an altered estradiol-induced LH surge. The experimental approach used allowed for a cell-selective, temporally-controlled suppression of ER-alpha expression, providing further evidence of the critical role of RP3V kisspeptin neurons in the estrogen positive-feedback effect. The preovulatory LH surge is a variable phenomenon and is better evaluated using serial blood sampling. Although the assessment of the estradiol-induced LH surge was performed in one terminal blood collection, c-Fos expression in GnRH neurons was used as a reliable proxy of the LH surge occurrence. The present findings also suggest that GABA neurotransmission in the preoptic area itself is not involved in the positive-feedback effect of estradiol on LH secretion.

---

## [Referee Report · Reviewer #3 (Public Review)]

Summary: The present study sought to investigate the role ERα expressed in Gabaergic neurons of the rostral periventricular aspect of the third ventricle (RP3V) and medial preoptic nucleus (MPN) in the positive feedback using genetically driven Crispr-Cas9 mediated knockdown of ESR1 in VGAT expressing neurons. ESR1 Knockdown in preoptic gabaergic neurons led to an absence of LH surge and acyclicity when associated with severely reduced kisspeptin (Kp) expression suggesting that a subpopulation of neurons co-expressing Kp and VGAT are key for LH surge since total absence of Kp is associated with an absence of GnRH neuron activation and reduced LH surge. Although the implication of kisspeptin neurons was highly suspected already, the novelty of these results lies in the fact that estrogen signaling is necessary in only a selected fraction of them to maintain both regular cycles and LH surge capacity.

Strengths:

Remarkable aspects of this study are, its dataset which allowed them to segregate animals based on distinct neuronal phenotype matching specific physiological outcomes, the transparency in reporting the results (e.g. all statistical values being reported, all grouping variables being clearly defined, clarity about animals that were excluded and why) and the clarity of the writing. Another remarkable feature of this work lies in the analysis of the dataset. As opposed to the cre-lox approach which theoretically allows for the complete ablation of specific neuronal populations, but may lack specificity regarding timing of action and location, genetically driven in vivo Crispr-Cas9 editing offers both temporal and neuroanatomic selectivity but cannot achieve a complete knock down. This approach based on stereotaxic delivery of the AAV encoded guide RNAs comes with inevitable variability in the location where gene knockdown is achieved. By adjusting their original grouping of the animals based on the evaluation of the extent of kisspeptin expression in the target region, the authors obtained a much clearer and interpretable picture. Although only few animals (n=4) displayed absent kisspeptin expression, the convergence of observations suggesting a central impairment of the reproductive axis is convincing. Finally, the observation that the pulsatile secretion of LH is maintained in the absence of Kp expression in the RP3V lends support to the notion that LH surge and pulsatility are regulated independently by distinct neuronal populations, a model put forward by corresponding author a few years ago.

---

## [Author Response]

The following is the authors’ response to the original reviews.

**Reviewer #1 (Recommendations For The Authors):**
Weaknesses: One minor weakness in this study is the conclusion that the guide RNAs didn't seem to have unique effects on GnRH cFos expression or the reproductive phenotypes. Though the data indicate a 60-70% knockdown for both gRNA2 and gRNA3, 3 of the 4 gRNA2 mice had no cFos expression in GnRH neurons during the time of the LH surge, whereas all mice receiving gRNA3 had at least some cFos/GnRH co-expression. In addition, when mice were re-categorized based on reduction (>75%) in kisspeptin expression, most of the mice in the unilateral or bilateral groups received gRNA2, whereas many of the mice that received gRNA3 were in the "normal" group with no disruption in kisspeptin expression. Thus, additional experiments with increased sample sizes are needed, even if the efficacy of the ESR1 knockdown was comparable before concluding these 2 gRNAs don't result in unique reproductive effects.

Response: A draw back of the CRISPR approach is the substantial mosaicism in gene knockdown that is unavoidable due to the nature of DNA repair in each cell relying on several competing pathways. As such, variable knockdown occurs in each mouse as shown in Fig.1C. In the case of the correlation between RP3V ESR1 knockdown and cFos in GnRH neurons (Fig.4C), three gRNA3 and four 4 gRNA2 mice look to be very similar with two gRNA3 mice having knockdown but normal cFos activation. The reasons for this are not known and it is very likely chance that these two (of nine) mice happened to have received gRNA3. This issue becomes exacerbated when animal group numbers unintentionally become smaller with the re-grouping on the basis of kisspeptin expression. The key point here is that each “kisspeptin grouping” remains mixed in terms of gRNA2 and gRNA3 mice so that gRNA3 mice did contribute to the “bilateral group” even if it was only one of four mice. The practicalities of repeating this work are substantial and we do not think justified. We would note that we have previously used Kiss-Cre mice to undertake CRISPR knockdown of ESR1 in RP3V kisspeptin neurons but this failed to target sufficient cells with Cas9 to be experimentally useful.

In Figure 2B (gRNA2), there appear to be 4 mice (4 lines) that have a normal cycle length and then drop to 0 for the cycle length. However, in the Figure legend, it states that there were 3 gRNA2 mice that had a cycle length of 0. Can the authors clarify if it was 4 mice (as indicated in Figure 2B) or 3 mice (as indicated in the legend) that received gRNA2 and exhibited constant estrus?

Response: We have now clarified in the text that 3 gRNA2 mice went into constant estrus, the other mouse was in constant diestrus, also scored as “0” cycles.

In Figure 3H, there is one green data point that has an LH level of around 0.15 and % VGAT with ESR1 around 10%. However, that data point does not appear in Figures 3I and 3J, when you would expect it to be in a similar place (~10%) on the x-axis in those Figures. Was it excluded? If so, please elaborate on the justification for excluding that data point.Response: This was one of the three mice that exhibited no LH pulses so we were only able to report on mean LH levels.Similarly, in Figure 3K, there is a blue data point that is almost at 0 for both the x-axis and the y-axis. However, that data point does not show up in Figures 3L and 3M around 0 on the x-axis as you would expect. Can the authors clarify where this data point went in Figures 3L and 3M?

Response: This was one of the three mice that exhibited no LH pulses so we were only able to report on mean LH levels.

**Reviewer #2 (Recommendations For The Authors):**
Finally, the study leaves unanswered the role of GABA itself. As there was no evident phenotype for the ESR1 knockdown in GABA neurons that do not coexpress kisspeptin, this suggests that GABA neurotransmission in the preoptic area is not involved in the estrogen regulation of LH secretion.

Response: The current evidence for no substantial role of GABA from RP3V neurons in the LH surge agrees with our prior in vivo work showing that low frequency optogenetic stimulation of RP3V kisspeptin neurons (only GABA release) has no impact on LH secretion (doi: 10.1523/JNEUROSCI.0658-18.2018).

1. Title. The present data do not clearly demonstrate the blockade of the LH surge. Thus, the statement that "abolishes the preovulatory surge" is an overinterpretation of the findings.

Response: We agree and now use “suppresses the preovulatory surge”.

1. Fig. 3. The numbers of individual data points per group change for the different LH pulse parameters, but they should not (Fig. 3 E-G).

Response: This occurs because one mouse in each group had no LH pulses so that only a mean value was available for these mice.

1. Fig. 4. (4B) The use of only one terminal blood collection (4B) is insufficient to comprehensively characterize the LH surge. It is not possible to conclude what was the actual effect on the LH surge, whether a blockade or altered amplitude or timing. Serial blood samples at 30- or 60-minute intervals should be used. For comparative purposes, the pulsatile LH secretion, which does not seem to be a major outcome in the study, was fully characterized (Fig. 3). (4C) The linear correlation between c-Fos/GnRH and RP3V/ESR1 appears to be well-fitted for gRNA2 (blue) but not gRNA3 (green). Although this is interpreted as an important result of the study, its description and consistency are not so clear. Authors should perform an Anova/ Kruskal-Wallis analysis of these data as a column graph (as in Fig. 4A, B) and discuss the discrepancies between gRNA2 and gRNA3.

Response: As noted in the manuscript, we agree that a single point LH measurement is a relatively inaccurate assessment of the LH surge and very likely underlies much of the substantial variability between mice. However, the extended duration of cFos expression in GnRH neurons at the time of the surge is a much more accurate “single point” indicator and we feel that these results better reflect the state of surge activation. This was noted in the original manuscript.

The linear correlations for the different preoptic regions are undertaken on the complete data set not on individual gRNA groups due to low N numbers in the sub-divided groups. However, column graphs of the RP3V and MPN look the same as Fig.4A and would not change the current interpretation. Please see comments to Reviewer 1 on discrepancies between gRNA2 and 3.

1. Table. It is unclear why the % VGAT with ESR1 was not statistically reduced in the "bilateral" animals. Would this mean that the ESR1 knockdown was not effective in this subgroup with the more consistent effects?

Response: Yes, this would be a reasonable interpretation suggesting that mice with kisspeptin ablation may have had a slightly different overall impact on ESR1 in VGAT neurons. However, this was not discernable from examining the anatomical distribution of AAV.

1. Discussion 1st paragraph. It is interpreted that mice lacking kisspeptin expression "failed to exhibit an LH surge". This should be revised.

Response: We believe that this is a correct statement. Mice lacking kisspeptin had LH surge values between 0.8 and 2.1 ng/ml that we would not consider consistent with being a surge.

1. Immunohistochemistry. It is not clear in the text how a cross-reaction between goat antirabbit 568 (ERa) and goat antirabbit/streptavidin 647 (mChery) was avoided when used in the same reaction.

Response: We were forced into this option due to the lack of different primary antisera to ESR1 and mCherry. We first stained for rabbit ESR1 detected by biotin anti-rabbit/ strep647 which resulted in confined nuclear staining (pseudo-blue; far red). The subsequent staining for rabbit mCherry was detected by goat anti-rabbit 568 that will indeed cross-react by binding to any free epitopes on the rabbit ESR1 primary antibody. However, this would not compromise interpretation as additional 568 labelling to the nucleus is essentially irrelevant when examining far red 647 nm emission and only mCherry cytoplasmic immunoreactivity was used to define the anatomical locations of the AAV spread. This is now clearly explained in the Methods section.

1. Statistical analysis. It is unclear when repeated measures Wilcoxon tests were used in the manuscript.

Response: Thank you for pointing this out. Only Wilcoxon paired test were used. Amended.

1. Data Availability. Further reference to supplementary information files was not found in the manuscript.

Response: A supplementary file with individual data for each mouse is now attached.

**Reviewer #3 (Recommendations For The Authors):**
Weaknesses:One aspect for which I have ambiguous feelings is the minimal level of detail regarding the HPG axis and its regulation by estrogens. This limited amount of detail allows for an easy read with the well-articulated introduction quickly presenting the framework of the study. Although not presenting the axis itself nor mentioning the position of GnRH neurons in this axis or its lack of ERα expression is not detrimental to the understanding of the study, presenting at least the position of GnRH neurons in the axis and their critical role for fertility would likely broaden the impact of this work beyond a rather specialist audience.

Response: We agree that this would provide a more complete picture and have modified the Introduction.

The expression of kisspeptin constitutes a key element for the analysis and conclusion of the present work. However, the quality of the kisspeptin immunostaining seems suboptimal based on the representative images. The staining primarily consists of light punctuated structures and it is very difficult to delineate cytoplasmic immunoreactive material defining the shape of neurons in LacZ animals. For some of the cells marked by an arrow, it is also sometimes difficult to determine whether the staining for ESR1 and Kp are in the same focal plane and thus belong to the same neurons. Although this co-expression is not critical for the conclusions of the study, this begs the question of whether Kp expression was determined directly at the microscope (where the focal plan can be adjusted) or on the picture (without possible focal adjustment). Moreover, in the representative image of Kp loss, several nuclei stained for fos (black) show superimposed brown staining looking like a dense nucleus (but smaller than an actual nucleus). This suggests some sort of condensed accumulation of Kp immunoproduct in the nucleus which is not commented. Given the critical importance of this reported change in Kp expression for the interpretation of the present results, it is important to provide strong evidence of the quality/nature of this staining and its analysis which may help interpret the observed functional phenotype.

Response: The kisspeptin immunoreactivity represents both fiber and cytoplasmic staining that can be difficult to discern in some cases. The reviewer can be assured that all counts were undertaken “live” on the microscope so that the plane of focus was adjusted to establish co-labelling. Please note that the nuclear immunoreactivity is for ESR1 and not cFos. Regardless, we struggle to see condensed brown staining over the black nuclei as suggested by the Reviewer. The kisspeptin staining is light brown and confined to just a few fibers in Fig.5B.

As acknowledged in the introduction, this study is not the first to use in vivo Crisp-Cas editing to demonstrate the role of kisspeptin neurons in the control of positive feedback. Although the present work achieved this indirectly by targeting VGAT neurons, I was surprised that the paper did not include more comparison of their results with those of Wang et al., 2019. In particular, why was the present approach more successful in achieving both lack of surge and complete acyclicity?

Response: Wang et al., reported an ~60% reduction in ESR1 expression in Kiss1-Cre (Elias) driven Cas9-expressing cells in the AVPV. As they did not examine kisspeptin expression itself it is unknown to what degree their editing impacted upon kisspeptin neurons. The other differentiating factor was that Wang focussed on the AVPV that only contains a minority of the preoptic kisspeptin population whereas we targeted the AVPV and PeN together. Thus, we suspect that the Wang phenotype arises from insufficient ESR1 knockdown in just the AVPV sub-population of preoptic kisspeptin neurons. We have added a comment to the Discussion as requested.

Moreover, why is it that targeting ESR1 in a selected fraction of GABAergic neurons can lead to a near-complete absence of Kp expression in this region? This is briefly discussed in the penultimate paragraph but mostly focuses on the non-kisspeptinergic GABA neurons rather than those co-expressing the two markers.

Response: We have modified this section to try and make it clear that it is very likely that all RP3V kisspeptin neurons would have been targeted to express Cas9 in this mouse model. Our very recent unpublished RNA scope data show that >80% of RP3V kisspeptin neurons express Vgat mRNA in adult mice.

Unless I have missed it, the target sequence of the guide RNAs is not mentioned. For reproducibility purposes and to allow comparison with Wang et al., 2019, this information should be provided.

Response: The target sequences for gRNA2 and gRNA3 were around exon 3 and are provided in the Supplementary files of McQuillan et al., 2022 (https://doi.org/10.1038/s41467-022-35243-z). The Wang et al study used the unusual strategy of designing sense and antisense gRNAs against the same sequence in Exon1.

The first result section is devoted to the design and validation of the guide RNA reports data that were recently published (McQuillan et al., 2022). It is actually acknowledged that the design was reported previously but as written it is not clear whether the actual validation was already reported. This should be said more clearly.

Response: Clarified as requested.

What was the rationale for choosing gRNA 2 and 3 and not 3 and 6 like in the McQuillan study?

Response: As all three gRNAs worked equally well, the choice of 2 and 3 was entirely pragmatic and only based upon quantities of packaged AAVs that we had produced and were available at the time.

Introduction, 4th paragraph: It would be clearer if GABAa receptor dynamics was replaced by GABAa receptors mediated neurotransmission or any other verbiage avoiding possible confusion with receptor mobility.

Response: Clarified as requested.

The section reporting the location of ESR1 knockdown is really clear about the number of animals included in the functional analyses. This is less clear for the number of mice involved in the evaluation of the extent of ESR1 knockdown in the previous section. Specifically, the text reports that 8 and 9 mice received gRNA3 in PVpo and MPN respectively, but the figure shows 7 and 8. This is likely explained by the mouse that was excluded due to normal ESR1 despite the correct positioning of the injection site. It is thus unclear whether this mouse was included in the calculation of the mean percentage of neurons reported in the previous page. Logically, this mouse should have been removed from this analysis and it is assumed that the sample size reported in the text is incorrect.

Response: thank you for picking this up - you are correct. In reviewing this point we realized that the gRNA-lacZ RP3V N numbers also were incorrect and have re-analyzed the data set completely resulting in even stronger significance levels.

In the section « CRISPR knockdown ESR1 in RP3V GABA-kisspeptin neurons », the extent of ESR1 knockdown is expressed in a counterintuitive manner as « <20% » which is thought to represent the percentage of cells expressing ESR1 rather than the actual knockdown (>80%). This should be clarified.

Response: Corrected as noted.

Page 6, 3rd line before the last paragraph, there is a mismatch between the highest p value reported in the text (0.242) and the value reported in the table (0.0242).

Response: Corrected thank you.

Similar to presenting F values for ANOVAs, H values should also be presented for Kruskal Wallis tests.

Response: Values have been added.

Immunohistochemistry : Origin and reference numbers of all primary antibodies should be reported as well as citation of studies where they have been validated. Although these protocols are standard, information regarding the duration of incubation is necessary to allow replication or for comparison purposes.

Response: We have included the RRID numbers for each of these antisera and added information on incubation times.

The section on data availability mentions the existence of supplementary files, but I see none.

Response: These have now been attached.

There are several typos or redundancies to be corrected. Here are a few examples but the manuscript should be carefully double-checked.Introduction, 3rd paragraph, line 4: upregulatedIntroduction, 4th paragraph, 4th line: « to » or « through » not both.Page 7, line 11 : KruskalPage 7, 6th line to the end: does this indicate 'the' general utility?Page 8, 2nd paragraph, line 13: Crispr

Response: Thank you for these edits.